# Impact of Perfluoro and Alkylphosphonic Self-Assembled Monolayers on Tribological and Antimicrobial Properties of Ti-DLC Coatings

**DOI:** 10.3390/ma12152365

**Published:** 2019-07-25

**Authors:** Michal Cichomski, Milena Prowizor, Ewelina Borkowska, Ireneusz Piwoński, Anna Jędrzejczak, Mariusz Dudek, Damian Batory, Natalia Wrońska, Katarzyna Lisowska

**Affiliations:** 1Department of Materials Technology and Chemistry, Faculty of Chemistry, University of Lodz, Pomorska 163, 90-236 Lodz, Poland; 2Institute of Materials Science and Engineering, Lodz University of Technology, Stefanowskiego St 1/15, 90-924 Lodz, Poland; 3Department of Vehicles and Fundamentals of Machine Design, Lodz University of Technology, Stefanowskiego 1/15, 90-924 Lodz, Poland; 4Department of Industrial Microbiology and Biotechnology, Faculty of Biology and Environmental Protection, University of Lodz, Banacha 12/16, 90-237 Lodz, Poland

**Keywords:** titanium incorporated diamond-like carbon, self-assembled monolayers wettability, Nano-/microtribology, antimicrobial activity

## Abstract

The diamond-like carbon (DLC) coatings containing 1.6%, 5.3% and 9.4 at.% of Ti deposited by the radio frequency plasma enhanced chemical vapor deposition (RF PECVD) method on the silicon substrate were modified by *n*-decylphosphonic acid (DP) and 1H, 1H, 2H and 2H-perfluorodecylphosphonic acid (PFDP). The presence of perfluoro and alkylphosphonic self-assembled monolayers prepared by the liquid phase deposition (LPD) technique was confirmed by Fourier transform infrared spectroscopy (FTIR). It was shown that DP and PFDP monolayers on the surface of titanium incorporated diamond-like carbon (Ti-DLC) coatings had a huge influence on their wettability, friction properties, stability under phosphate- and tris-buffered saline solutions and on antimicrobial activity. It was also found that the dispersive component of surface free energy (SFE) had a significant influence on the value of the friction coefficient and the percentage value of the growth inhibition of bacteria. The dispersive component of SFE caused a reduction in the growth of bacteria and the friction coefficient in mili- and nano-newton load range. Additionally, both self-assembled monolayers prepared on Ti-DLC coatings strongly reduced bacterial activity by up to 95% compared to the control sample.

## 1. Introduction

Carbon-based coatings have enjoyed growing interest and can be successfully used in the electronics and microelectronics, especially in micro/nanoelectromechanical systems (MEMS/NEMS) [1,2]. Due to the high biocompatibility, chemical inertness, corrosion resistance and low friction coefficient of the diamond-like carbon (DLC) coatings, they are also ideal for biomedical and tribological applications [3,4,5,6]. The possibility of using DLC is wide but the high internal stresses occurring in the coating is a major problem that may result in poor adhesion to the substrate and a tendency to delamination. The use of dopants in DLC structure is the most effective method for solving this problem [7,8,9]. It is important to find a dopant that allows reducing the internal stresses while maintaining the existing properties of the DLC film [10]. The radio frequency plasma enhanced chemical vapor deposition (RF PECVD) method is one the most important techniques that allow for deposition of good quality pure and doped DLC coatings. The method is easy to implement in the industry and allows the use of different precursors containing dopant atoms (e.g., titanium) to improve mechanical properties [2,11,12].

In addition, titanium and titanium dioxide are characterized by good biocompatibility, chemical stability and corrosion resistance. It was demonstrated, that DLC films containing titanium or titanium oxide showed good biocompatibility towards osteoblasts. A number of properties make them a strong candidate to be used as a matrix, that embeds different compounds with a potential antimicrobial effect. Moreover, titanium nanoparticles characterized by activity against microorganisms, including Gram positive bacteria [13,14,15,16], Gram negative bacteria [16,17] and fungi [18]. Unfortunately, the use of doped DLC coatings in tribology is limited in view of the influence of humidity on an increase of the friction coefficient. The solution to this problem is the necessity for an ultrathin lubricant film formed by compounds creating self-assembly monolayers (SAMs). These hydrophobic surfaces characterized by a high contact angle and low contact angle hysteresis are extremely important in liquid flow applications because they show low adhesion and drag reduction [19,20]. Therefore, another good way to increase hydrophobic properties of the Ti-DLC is to produce SAMs on their surface. An important class of self-assembled organic molecules is fluoroalkylchlorosilanes and alkylphosphonic acids [21,22]. Silanes are usually the most frequently used to create a surface with hydrophobic properties and are widely described in the literature [23]. Alkylphosphonic acids represent an unappreciated group of compounds with expectation for the medicine [24]. The most popular derivative of alkylphosphonic acid is phosphomycin, which interferes with cell wall synthesis by inactivating phosphoenolpyruvate synthetase [25]. Mixtures of phosphonopeptides are also considered as strong antimicrobial agents [26]. Alkylphosphonic acids form well ordered, densely packed and stable SAMs, which are extremely important for tribological applications [27]. Moreover, an undoubted advantage of alkylphosphonic acids over silane compounds is their non-toxicity. Adsorbed molecules of alkylphosponic acid can be used to cover surfaces with a self-cleaning ability, micro/nanochannels, stents, endoprosthesis elements or other elements for biomedical applications. An element of novelty in this work is the demonstration of the inhibition of the growth of bacteria and also a reduction of coefficient of friction via perfluoro and alkylphosphonic self-assembled layers. These compounds were also chosen to compare the effect of their structure on physicochemical properties of Ti-DLC. In order to understand these phenomena, we presented the influence of the acid-base and dispersive components of surface free energy (SFE) on the friction coefficient and on the antibacterial activity, which no research was reported before.

In this research we are going to show that the nature of the chemical structure of investigated compounds has a huge influence on wettability, friction properties, stability under phosphate- and tris-buffered saline solutions and antimicrobial activity. It was found also that the dispersive component of SFE contributes to the inhibition of bacterial growth and reduces the coefficient of friction on both micro and nano scales.

## 2. Experimental

### 2.1. Ti-DLC Coating Deposition Process 

Titanium-containing DLC coatings with the thickness of 100 ± 2 nm were deposited on silicon substrates Si(100) using the radio frequency plasma enhanced chemical vapor deposition (RF PECVD) method at 400 V of negative self-bias under 20 Pa pressure of mixture of methane (CH_4_) and titanium (IV) isopropoxide (Ti[OCH(CH_3_)_2_]_4_). Controlling a content of precursors in work atmosphere pure DLC and three Ti-DLC structures with different contents of titanium (1.6 at.%—Ti-DLC1, 5.3 at.%—Ti-DLC2 and 9.4 at.%—Ti-DLC3) were deposited. The details of the deposition process can be found elsewhere [28]. The thickness was controlled by choosing an appropriate duration of the deposition process. Additionally, the thickness of the manufactured Ti-DLC coatings was measured with the use of field emission scanning electron microscope (FE-SEM) NovaNanoSEM 450 (FEI) equipped with a Schottky gun. Data was collected at an accelerating voltage of 5 kV. 

### 2.2. Formation of Perfluoro and Alkylphosphonic Self-Assembled Monolayers

Self-assembled layers of perfluoro and alkylphosphonic were prepared on the Ti-DLC coatings with different contents of Ti by use of the liquid phase deposition (LPD) method. The modifications were performed with the use of *n*-decylphosphonic acid (DP) and 1H, 1H, 2H and 2H-perfluorodecylphosphonic acid (PFDP), which were purchased from ABCR, GmbH & Co. KG, Karlsruhe, Germany. Prior to the modification the samples were subjected to low pressure air plasma (Diener Electronic Plasma-Surface-Technology, Zepto, 40 Hz, 100 W) in order to remove organic contaminants but also to initiate the formation of –OH and Ti–O–Ti groups. These groups play the role of anchoring centers for modifying compounds [29].

DP and PFDP solutions were prepared by dissolving the modifier powder in ethanol at room temperature and under ambient conditions. The concentration of PFDP and DP solutions was 0.5% and 0.05% respectively. These concentrations were selected after the previous optimization performed. In the next step, the Ti-DLC with perfluoro and alkylphosphonic layers were removed from the acid solution and rinsed in ethanol. Finally, the samples after the deposition process were heated at 50 °C for 24 h.

### 2.3. Surface Characterization

Measurements of water contact angle and the quasi-static contact angle were employed to evaluate the wettability of pure Ti-DLC before and after the modification. The DSA-25 Drop Shape Analysis System (KRÜSS GmbH, Hamburg, Germany) working at 22 ± 2 °C and 45% ± 5% humidity was used for the measurements. The measurement included the placement of three types of liquids (water, diiodomethane and glycerine) on each surface at five different locations and measuring the wetting angle for these liquids. Knowing the values of contact angles, a surface free energy was calculated by the Van Oss–Chaudhury–Good method [30]. The quasi-static contact angle and contact angle hysteresis were measured by the sessile drop. The drops having a volume of 1.5 to 3 μL were dispensed automatically with a microsyringe. The final results were obtained using the software for the automatic measurement of advancing and receding contact angles.

The effectiveness of carried modification was investigated using a Nicolet iS50 spectrometer equipped with a GATR accessory from Harrick Scientific Products Inc. The ultra-high sensitive, low noise, linearized MCT (cooled with liquid nitrogen) detector was used. In the case of the present investigations, FTIR measurements were performed in the spectral range of 700–3100 cm^−1^. All spectra were recorded by collecting 64 scans at a 4 cm^−1^ resolution, in dry air atmosphere. 

The Solver P47 Atomic Force Microscopy(AFM) apparatus was used to study the morphology, roughness and friction coefficient of the coatings in nanoscale. All measurements were carried out in air under ambient conditions (20 ± 2 °C and 30% ± 2% humidity). The topography images were recorded employing the tapping mode. The scanned area was 2 μm × 2 μm at the scan rate 0.5 Hz. The values of the friction coefficient were calculated from the slope of the friction force versus normal force plots. During the measurements, the following parameters were used: Applied loads ranged from 5 to 100 nN, scan rate of 1 Hz and scan size of 1 μm × 1 μm. The obtained average data of measurements from three different places for each coating are shown in the graph.

Tribological tests were carried out using a reciprocating ball-on-flat T-23 microtribometer with the following parameters: Velocity of 25 mm/min, traveling distance of 5 mm, range of load from 30 to 80 mN, humidity (30% ± 2%) and temperature (20 ± 2 °C). Si_3_N_4_ sphere having 5 mm diameter and average roughness of 5.5 ± 0.5 nm was used as a counterpart. The measurements were performed in three different locations of all surfaces and were repeated three times.

The stability of pure and modified Ti-DLC coatings was tested by immersing the samples in tris-buffered saline (TBS) and phosphate buffered saline (PBS) [31]. The samples were exposed to the solutions for different time (from 0.25 to 720 h). After exposure to the solutions, the water contact angle on each surface was examined.

### 2.4. Determination of the Antimicrobial Activity of Analyzed Samples

The antibacterial activity of perfluoro and alkylphosphonic acids were tested in the solution against *Staphylococcus aureus* ATCC 6538 and *Escherichia coli* ATCC 25922 using a modified broth microdilution method, according to the recommendation of the Clinical Laboratory Standard Institute (CLSI M07-A8). In the experiments the Mueller–Hinton broth was used, the final optical density was about 5 × 10^5^ colony forming units (CFU). The self-assembled compounds were dissolved in sterile deionized water and were tested in concentrations ranging from 0.1 µg/mL to 200 µg/mL. The obtained data were compared to the control of biotic samples without the organic compounds.

Next, the antimicrobial activity of perfluoro and alkylphosphonic self-assembled layers on Ti-DLC was examined. The antibacterial activity was tested against *Staphylococcus aureus* ATCC 6538 and *Escherichia coli* ATCC 25922 as in previous works [32,33] using the Japanese Industrial Standard JIS Z 2801:2000. Bacteria were cultured on Luria Bertani (LB) medium at 37 °C on a rotary shaker. After the incubation, the test inoculum of *S. aureus* and *E. coli*, containing 1 × 10^5^ colony forming units (CFU per mL) in 500-fold diluted LB medium was prepared. Next, the bacteria suspension was applied to tested coatings of 1 cm × 1 cm. Diamond-like carbon (DLC) coatings were analyzed as a control sample. After dripping the suspension of selected bacteria on the coatings, each sample was covered with a sterile film. Then, the samples were incubated in the moist chamber in the dark for 24 h at 37 °C. After incubation, the samples were put in the sterile tube containing phosphate buffer and vortexed. After that, coatings and films were removed from the tubes and with the remaining solution, a serial dilution was performed in phosphate buffer. Out of each dilution, 100 µL of bacterial suspension was seeded on agar plates and incubated for 24 h at 37 °C. Next, the viable cells of *S. aureus* or *E. coli* were counted. Each type of coatings or solutions was tested in triplicate and analyzed individually in three independent experiments. The antibacterial activity of the tested coatings was calculated as the percentage of bacterial growth inhibition (+/− SD) toward the control sample without perfluoro and alkylphosphonic layers. 

## 3. Results and Discussion

The chemical structure and the presence of self-assembled molecules on Ti-DLC were confirmed with the use of FTIR spectroscopy (Figure 1). FTIR spectra for Ti-DLC appeared to reveal signals mainly in the range for Ti-OH bonds (at about 1455 cm^−1^), Ti-O-Ti bands (at about 700 and 820 cm^−1^) and Ti-O bonds (at about 940 cm^−1^). Observed bond vibrations indicate the formation of titanium oxides network, which is higher with the increasing participation of admixture in the DLC structure. This was confirmed by increasing intensity of these bands with the increasing titanium concentration. It also indicates that both titanium and oxygen are incorporated in DLC. Another bands characteristic for these surfaces were Ti-CH_3_ bonds (at about 1255 and 1305 cm^−1^). The intensity of absorbance of these bands constantly increased with amounts of incorporated titanium. This fact confirms that titanium was connected to a carbon atom and played a dominant role in the tribological properties of these surfaces. 

On the spectra recorded after modification, phosphonic group characteristics appeared in the 700–1500 cm^−1^ spectral range. The peak at about 950 cm^−1^ was assigned to P–O–M bond vibrations. It was confirmed that the perfluoro and alkylphosphonic acid molecules reacted with the surface of the coating and formed a chemical bond. The P=O stretching vibration appeared in the region 1085–1415 cm^−1^, while the P–O vibration was noted at 972–1030 and 917–950 cm^−1^. The fact that the peak at about 1230 cm^−1^ corresponding to the P=O group was clearly visible on all IR spectrum could indicate a strong interaction between the phosphonic group and the surface. In the case of Ti-DLC modified by DP, the symmetric (*υ_s_* CH_2_) and asymmetric (*υ_a_* CH_2_) C–H stretching modes at 2850 and 2920 cm^−1^ corresponding to the methylene groups were visible in the IR spectra. In the same region, the symmetric (*υ_s_* CH_3_) and asymmetric (*υ_a_* CH_3_) C–H stretching modes were present at 2880 and 2960 cm^−1^, respectively. DP molecules had the methylene backbone group and the peak observed at 2958 cm^−1^ came from the front group (–CH_3_). The –CH_2_ groups occurring in the carbon chain of DP compound were also found at 2851 and 2920 cm^−1^. More –CH_2_ than –CH_3_ groups were present in its molecular structure, therefore a more intense peak was observed for the methylene bands than the methyl groups. These peaks were not present for Ti-DLC modified by PFDP while several peaks indicating the presence of fluoroalkyl groups in the sample in the range of 750–1300 cm^−1^ were observed. The bands noted at around 1200 cm^−1^, 1147 cm^−1^, 1114 cm^−1^ and 779 cm^−1^ corresponded to asymmetric and symmetric stretches of C–F for –CF_2_ and –CF_3_ groups originating from the fluoroalkyl chain. 

Perfluoro and alkylphosphonic layers present on Ti-DLC coatings with different concentrations of titanium gave them new, unique properties. Table 1 shows the results of the hydrophobicity analysis of Ti-DLC before and after DP and PFDP modification. It was found, that the titanium made the surface of Ti-DLC structures more hydrophilic, which was observed through decreasing the water contact angle value with the increasing amount of titanium in Ti-DLC. This fact confirms also the FTIR analysis in which it was shown that the presence of C–O and Ti–O bonds increased the hydrophilicity of the surface. Generally, the lowest value of the contact angle was received for Ti-DLC3 (Ti-DLC with 9.4 at.% of Ti).

The wettability of the studied surfaces was also measured by means of water contact angles (SCA) and the advancing (θ_a_), receding (θ_r_) contact angles and hysteresis (Δθ). Hysteresis cannot be eliminated completely, because it depends on many factors such as the adhesion hysteresis, surface roughness and inhomogeneity [34]. In our study, we could clearly see (Figure 2) that the change in the static and quasi-static contact angle values was dependent on the surface roughness expressed via the root mean square (RMS) surface roughness. Ti-DLC with 1.6 at.% of Ti exhibited generally lower values of contact angle hysteresis than Ti-DLC with a higher concentration of titanium, that was related with their high RMS values (0.33 ± 0.01 nm for Ti-DLC1, 0.31 ± 0.01 nm for Ti-DLC2 and 0.30 ± 0.02 for Ti-DLC3 coatings). Comparing these three types of Ti-DLC it was noticed that the surface of Ti-DLC3 coating was the flattest (Figure 2), which affects its hydrophilic properties.

After the surface modification, the water contact angle significantly increased, indicating that the surface hydrophobicity was improved. It is related to the presence of the well-ordered perfluoro and alkylphosphonic layer on the surface. The most hydrophobic properties showed the surface modified by PFDP. The –CF_3_ group present in perfluorinated self-assembled acid was more hydrophobic than the –CH_3_ group in decylphosphonic acid. It is associated with the appearance of fluorine in the compound structure [35]. However, there were clear differences in the hydrophobicity when the layer was formed of compounds containing the –CH_3_ and –CF_3_ functional groups. The size of the hydrogen and fluorine atoms played an important role. Fluorine was significantly larger than hydrogen so consequently, the average volumes of the CF_2_ and CF_3_ groups were estimated as 38 A^3^ and 92 A^3^, respectively compared to 27 A^3^ and 54 A^3^ for the CH_2_ and CH_3_ groups. It caused a greater stiffness of perfluorinated chains than their hydrocarbon equivalents, which prevented a drop of water to penetrate into the surface. Furthermore, effective overlapping of orbitals caused the C–F bond to be more stable (485 kJ mol^−1^) compared to a standard C–H bond (425 kJ mol^−1^). It meant that the dense electron cloud of the fluorine atoms acted as a cover that protects the perfluorinated chain against the approach of water molecules [36,37,38]. This was confirmed by the obtained values of contact angle and SFE. Comparing the non-polar –CF_3_ and –CH_3_ groups, better hydrophobic properties were characteristic for CF_3_ groups. Describing the hydrophobic properties, it is particularly important to take into consideration the roughness of the studied samples. After the deposition process, the RMS values for all coatings increased. The highest increase of RMS value was observed after the formation of PFDP on Ti-DLC1 coating (1.41 ± 0.02 nm). In the case of modification by non-fluorinated DP the registered value was 0.75 ± 0.02 nm. These facts caused the high hydrophobicity of perfluorinated layers.

Comparing the advancing contact angles of pure and modified coatings the significant difference in their values in favor of coatings with self-assembled compounds can be seen. The highest value of the advancing contact angle was obtained for the coating with the lowest content of titanium and with the phosphonic self-assembled layer. What is more, a similar trend also occurred in the case of a receding contact angle. After modification, the highest receding contact angle also was obtained for the DLC with 1.6 at.% of Ti and with DP and PFDP layers, reaching the values of 121° and 123° respectively. The high values of the advancing contact angle and the receding contact angle simultaneously caused a low hysteresis, which was 7.7° for Ti-DLC1/DP and 6.3° for Ti-DLC1/PFDP. For Ti-DLC2 and Ti-DLC3 modified by DP and PFDP, the values of the receding contact angle were in the range of 104–117° causing a higher hysteresis for these coatings compared to those described earlier. Very low contact angle hysteresis and high water contact angle testifies to the high surface hydrophobicity, so when the surface is characterized by low contact angle hysteresis the water behaves as a spherical droplet, with a low roll-off angle. This indicates that the highest value of the advancing contact angle was obtained for self-assembled monolayers, which are well ordered and the water molecule does not penetrate into the surface of Ti-DLC. 

For better understanding of the wetting mechanism of solid surfaces, the surface free energy (SFE) and the corresponding acid-base and dispersive components were investigated using the Van Oss–Good method. The Van Oss divided the total surface free energy (γ^TOT^) of a solid into two components, dispersive (γ^LW^) and acid-base (γ^AB^) component, presented by the equation:γ^TOT^ = γ^LW^ + γ^AB^.

The dispersive component γ^LW^ is based on Lifshitz–Van der Waals interactions and the acid-base component γ^AB^ is based on hydrogen bonding interactions. The acid-base component is the sum of electron donor component (γ_AB_^−^) and an electron acceptor component (γ_AB_^+^).
γ^AB^ = γ_AB_^+^ + γ_AB_^−^.

The SFE components, both dispersive and acid-base have an influence on wetting and tribological properties. Table 2 gives the SFE data, with the distinction between acid-base and dispersive components. It can be clearly seen that the SFE and acid-base component increased with the increasing concentration of titanium in the Ti-DLC coatings. It is related to the formation of titanium oxides on the surface. Titanium atoms present in the structure of Ti-DLC are free to bond with carbon atoms but also with oxygen and water, which are present during the deposition process as well as in the atmosphere. Therefore, as the concentration of titanium in the coating increased, the number of C–O and Ti–O polar bonds also increased, which in consequence increased the hydrophilicity of the Ti-DLC surface. 

The changes in the hydrophobicity of the modified Ti-DLC were also related to the acid-base and dispersive components of SFE as seen in Table 2. Generally, after the modification, the values of acid-base components of SFE changed considerably. In the case of Ti-DLC modified by DP, the surface energy value determined mainly the acid-base component, which was significantly reduced. In turn, the SFE value of Ti-DLC after PFDP deposition was influenced by both components of SFE: Acid-base and dispersive, which decreased by as much as about 50%–70% in relation to pure Ti-DLC. Table 2 also shows the values of the donor (γ_AB_^−^) and an acceptor (γ_AB_^+^) of the acid-base (γ^AB^) component. It was noticed that with the increasing concentration of titanium in the DLC structure, the γ_AB_^−^ surface energy component increased. While the donor component increased, the acceptor component of Ti-DLC coatings almost had not changed and was approximately zero. Similar results were obtained by Zhao et al. [39]. As can be seen the γ_AB_^−^ value decreased for the surface after DP and PFDP deposition in comparison with pure Ti-DLC. It is associated with the presence of monolayer where the hydrophobic part of molecules is oriented outwards. A similar behavior was observed by Yan et al. [40]. In addition, Zhao et al. [39] reported that if the electron donor component is large then the surface is more negatively charged. Therefore in the case of surface modification by DP and PFDP the high value of the donor component generates a negative charge on the surfaces. The contribution of the acceptor component is practically negligible. 

The terminal group of perfluoro and alkylphosphonic acid had also a significant influence on the tribological behavior of Ti-DLC coatings. Figure 3 shows the coefficient of friction measured using a microtribometer for all studied surfaces. The highest value of the advancing contact angle was obtained for the coating with the lowest content of titanium and with self-assembled monolayers adsorbed on the Ti-DLC, which caused a decrease in the coefficient of friction compared to the unmodified coatings—previously observed in macroscale [28]. In our study, the coefficient of friction obtained in the milinewton load range for pure Ti-DLC was the lowest for Ti-DLC with 1.6 at.% of Ti. Similarly to SFE, the coefficient of friction increased with an increasing amount of incorporated titanium. After modification by self-assembled compounds, the coefficient of friction was significantly reduced. A considerable reduction in the obtained values of the coefficient of friction between the unmodified and modified coatings was connected with hydrophobicity and low SFE of the modified surfaces. The reduction in the coefficient of friction values primarily resulted from the fact that the friction forces were dominated by surface interactions. Moreover, the reduction in the coefficient of friction also resulted from the presence of well-ordered layers on the coatings. Ti-DLC after modification regardless of the type of the modifier used exhibited lower values of coefficient of friction than their pure equivalent. It is connected with the presence of molecules of alkylphosphonic layers on the surface, which acted as a lubricant during the friction process. The PFDP was a better lubricant and exhibited lower values of the friction coefficient than DP, which was associated with the stiffness of the chains. The larger size of the fluorine atom compared to hydrogen caused the backbone structure rotation of fluorinated chains to be much lower than non-fluorinated ones due to steric effects. Therefore, the hydrophobic layer formed by fluorinated chains was more rigid than the layer created by alkylphosphonic acid molecules.

In nanoscale, similarly to the microscale, the coefficient of friction after modification by perfluoro and alkylphosphonic layers showed lower values compared to the pure coatings. This was due to the presence of a well-ordered layer on the surface of the coating, which exhibited completely different interactions with the surface of the counterpart than the pure Ti-DLC. As previously noted for pure coatings, the presence of polar bonds was responsible for an increase in capillary forces that had a large impact on the friction force and in consequence on the coefficient of friction. In the case of a hydrophobic surface, the capillary force was low, therefore the coefficient of friction was also decreased. Comparing the effect of the used modifier, the lowest values of the coefficient of friction were obtained for the Ti-DLC modified by the more hydrophobic PFDP layer. The obtained values of the coefficient of friction in the micro- and nanoscale for the same studied samples differed from each other and this was related to the size of the counterpart, different contact stresses, and load affecting the friction. In the case of friction measurements carried out in the microscale, the applied normal load was higher than in the nanoscale. What is important, in the case of friction measurements carried out in the nanonewton range of forces, we dealt with a point friction contact that was not present in the microscale. In nanoscale studies, the small apparent area of contact minimized the occurrence of an additional factor (plastic deformation) that increased the friction force. Therefore, the capillary forces and adhesional component had only an influence on the value of the friction force. The above mentioned factors had a significant influence on the recorded value of the friction force and consequently, on the differences in the friction coefficient values obtained in the nano- and microscale.

Additionally, the effect of the dispersive component of SFE in mili- and nanonewton load range for Ti-DLC structures on the friction coefficient value was observed, which can be seen also in Figure 3. In the case of pure Ti-DLC, the high value of the dispersive component was observed, which was reflected in the high values of the friction coefficient. After the formation of perfluoro and alkylphosphnic self-assembled monolayers on the Ti-DLC surface, it was found, that the frictional properties were effectively improved and the γ^LW^ components had been reduced. What is more, a drop in the dispersion component by as much as 90% resulted in a reduction of the friction coefficient up to a value of approximately 0.20. Generally, it was found that the value of the γ^LW^ surface energy component had a significant effect on the friction properties of Ti-DLC.

The water contact angle measurements were used in our studies, to characterize changes in perfluoro and alkylphosphonic layer stability after immersion in PBS and TBS solutions (Figure 4). In the case of pure Ti-DLC1, Ti-DLC2 and Ti-DLC3 coatings, no significant changes in the water contact angle were observed. The water contact angle persisted at a constant level regardless of the time of immersion. While for modified samples, these changes were observed. In the case of Ti-DLC1 modified by DP and PFDP immersed in PBS solution a significant decrease in the contact angle after 360 h from 123 ± 3° for DP and 127 ± 2° for PFDP to 97 ± 2° for DP and 100 ± 2° for PFDP was observed. These changes indicated that the self-assembled layer became less homogeneous and disordered or that the organic molecules were desorbed. A similar trend was observed for DP and PFDP immersed in TBS solutions. It also occurred in the case of modified Ti-DLC2 and Ti-DLC3.

Phosphonic self-assembled monolayers deposited on Ti-DLC not only showed high stability but also contributed to the increase of antibacterial properties of Ti-DLC coatings. In the literature, it was found that mixtures of phosphonopeptides are strong antimicrobial agents [26]. The obtained results showed that the tested solutions exhibited different activities against tested strains. DP as well as PFDP were more active against the Gram positive *Staphylococcus aureus* strain (Figure 5a) than against the Gram negative *Escherichia coli* strain. The addition of DP at a very low concentration of 0.1–10 µg/mL limited the growth of this strain almost by half, in comparison with control samples without self-assembled layers. At the highest concentrations (from 60 to 200 µg/mL), DP caused 70% growth reduction. The perfluoro alkylphosphonic acid exhibited better antimicrobial activity, in higher concentration (40 µg/mL). The Gram negative *E. coli* strain showed good tolerance to tested phosphonic acids compounds (Figure 5b), where the growth inhibition ranged from 10% to 30%. The best antibacterial effect against *E. coli* was obtained at the highest concentration of both compounds (from 120 to 200 µg/mL). A similar trend was observed by Abdelkader et al. who investigated the antimicrobial activity of alpha-aminophosphonic acids. The tested compounds inhibited the bacteria growth (Gram positive and Gram negative), with a better effect for Gram negative [41].

The aim of our study was also to determine the antimicrobial effect of perfluoro and alkylphosphonic layers formed on the surface of Ti-DLC with various contents of Ti. Generally, a smaller amount of bacteria occurred on the DLC coating with the highest content of Ti for both *S. aureus* and *E. coli* bacteria. This was due to the fact that the antimicrobial properties of pure Ti-DLC were affected by the acid-base component of SFE and more specifically by the donor interaction (γ^AB−^). As mentioned earlier, the higher the component value, the more negatively charged the surface. Therefore, the most negatively charged coating (Ti-DLC3) showed the best antimicrobial activity. The presence of bacteria on the surface was also related to adhesion interactions. To describe this phenomenon, the DLVO theory (Derjaguin–Landau–Verwey–Overbeek) could be used. According to this theory, the interaction between the surface and the bacteria is the sum of long-distance interactions: attracting (van der Waals) and repulsive (electrostatic). These forces determine the approach of the bacteria to the surface.

The dispersive component of SFE is related to the long-distance interactions. Figure 6 shows its influence on the growth of both *S. aureus* and *E. coli* bacteria. In all tested surfaces with self-assembled layers (DP, PFDP) a strong antibacterial activity against the Gram positive strain *S. aureus* (Figure 6b) was noticed. In the case of fluoride modified alkylphosphonic layers (PFDP) deposited on Ti-DLC, the growth inhibition was over 95% in all tested variants (compared to control). For PFDP layers the inhibition of *E. coli* growth reached a value of around 40% (Figure 6a). 

Summarizing, the dispersive component of Ti-DLC after modification had an influence on the inhibition of *E. coli* and *S. aureus* growth. When the value of this component decreased, the coatings exhibited better antimicrobial activity. This was particularly evident in the case of *S. aureus*. The growth of the Gram negative *E. coli* bacteria on the tested surfaces was higher than the Gram positive *S. aureus* bacteria and it was associated with the structure of the bacteria and its interaction with the coating.

## 4. Conclusions

The Ti-DLC coatings with different amount of incorporated Ti (1.6%, 5.3% and 9.4%) were prepared using the RF PECVD method. The self-assembled monolayers of perfluoro and alkylphosphonic acid by using the LPD method on the Ti-DLC surface were successfully deposited. The presence of self-assembled layers on the surface of the Ti-DLC was confirmed by using Fourier transform infrared spectroscopy. The chemical structure of investigated compounds had a huge influence on wettability, friction properties, stability under phosphate- and tris-buffered saline solutions and antimicrobial activity of examined self-assembled layers. The results of the static water contact angle, advancing and receding contact angle and also values of SFE indicated that the surface after deposition of perfluoro and alkylphosphonic layers changed its properties from hydrophilic to hydrophobic. The performed measurements suggested that the highest hydrophobic properties had a PFDP layer deposited on Ti-DLC with 1.6 at.% of Ti. The received value of the contact angle was 127.3°. What was more, the strong correlation between the dispersive component of SFE and both friction coefficient and antimicrobial activity was shown. The obtained results indicated that the increase in the dispersive component of SFE caused the increase of the coefficient of friction. The tribological measurements after the modification showed that the PFDP layers improved the friction properties and provided effective lubrication. The same trend was visible on a micro- and nanoscale. Stability tests confirmed that the Ti-DLC modified by self-assembled monolayers was stable in saline solutions for up to 30 days. An analysis of the antimicrobial properties of both self-assembled layers deposited on Ti-DLC showed an inhibition of bacteria growth by up to 95% for the Gram positive strain *S. aureus* and by about 40% for the Gram negative strain *E. coli*. For both cases the influence of the dispersive component as well as the SFE on the bacterial growth on the studied surfaces was shown. The obtained results indicated that Ti-DLC coatings modified by perfluoro and alkylphosphonic layers could be useful for potential biomedical applications.

## Figures and Tables

**Figure 1 materials-12-02365-f001:**
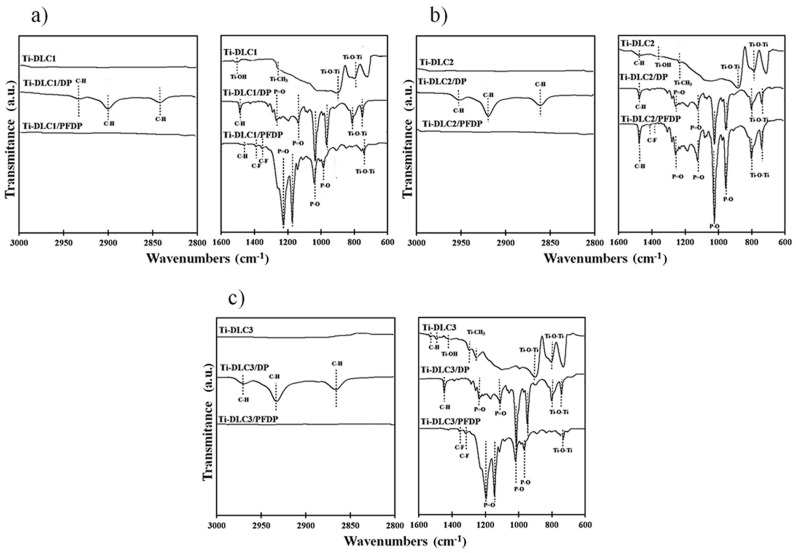
FTIR spectra of (**a**) Ti-DLC1, (**b**) Ti-DLC2 and (**c**) Ti-DLC3 coatings before and after modification by DP and PFDP.

**Figure 2 materials-12-02365-f002:**
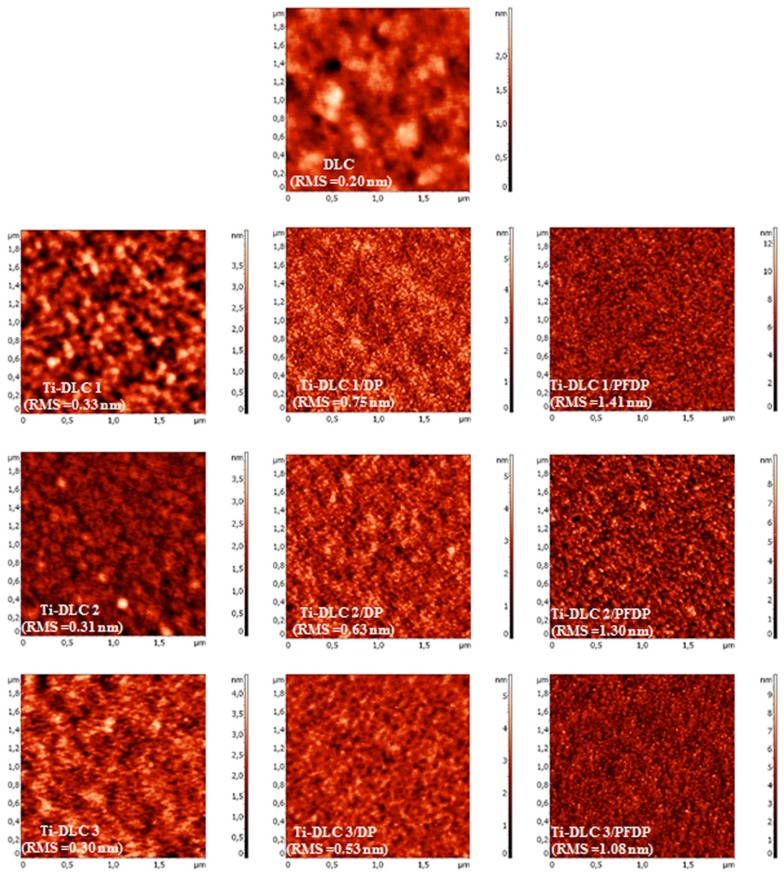
AFM Atomic Force Microscopy images of coatings with a different content of titanium before and after modification.

**Figure 3 materials-12-02365-f003:**
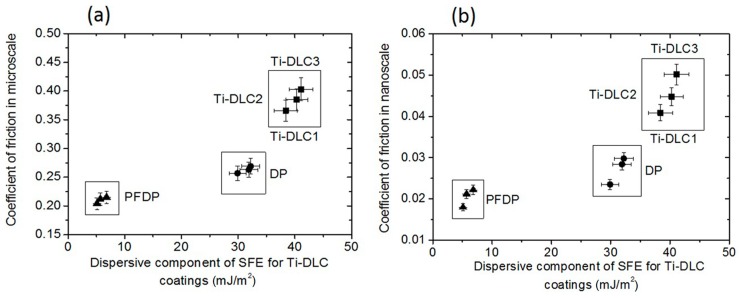
Effect of the dispersive component of surface free energy (SFE) on friction coefficient measured in the (**a**) micro- and (**b**) nanoscale.

**Figure 4 materials-12-02365-f004:**
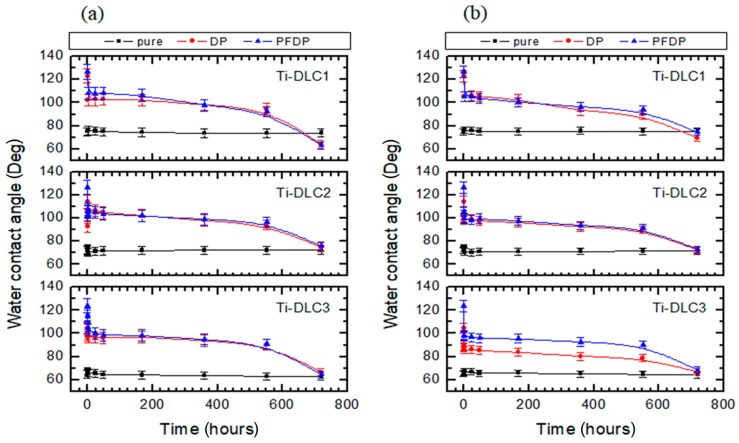
Stability of the water contact angle modified by DP and PFDP Ti-DLC after immersion in (**a**) PBS and (**b**) TBS solutions.

**Figure 5 materials-12-02365-f005:**
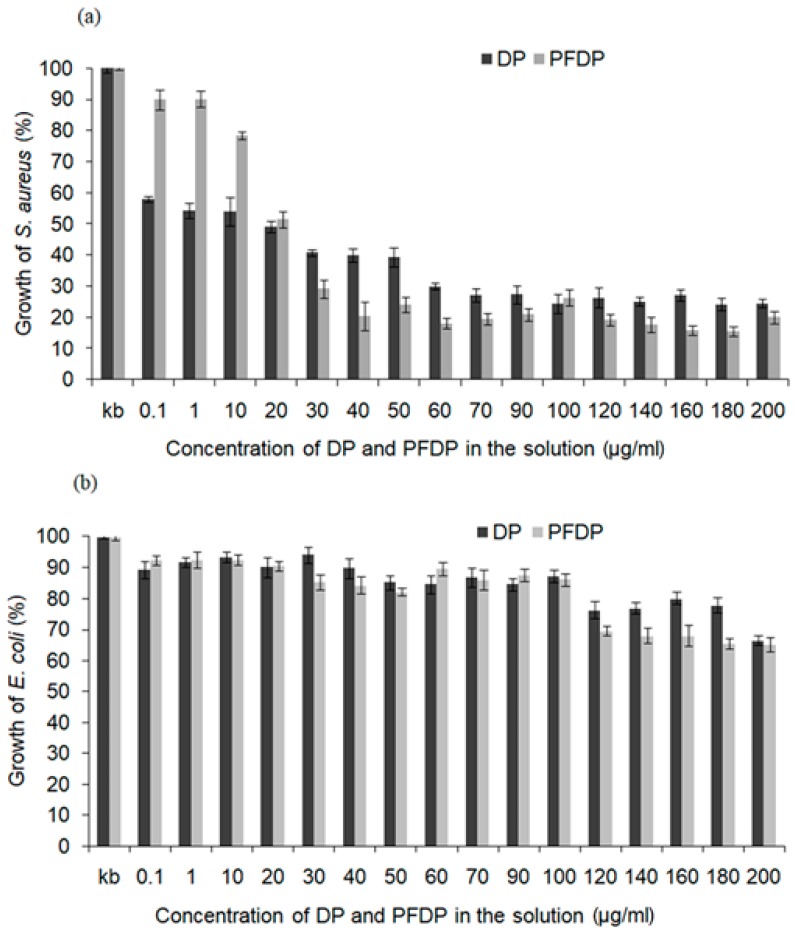
The growth of (**a**) *Staphylococcus aureus* and (**b**) *Escherichia coli* after 24 h of incubation with perfluoro and alkylphosphonic acids (DP, PFDP). Each bar represents the average and SD taken from *n* ≥ 3 wells from three independent experiments. The comparison was made using one-way analysis of the Student *t*-test. * *p* ˂ 0.05 vs. control group.

**Figure 6 materials-12-02365-f006:**
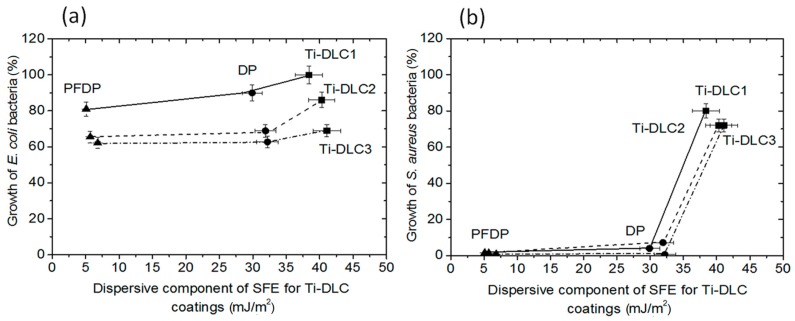
Effect of a dispersive component of SFE on the growth of (**a**) *E. coli* and (**b**) *S. aureus* after 24 h of incubation on perfluoro and alkylphosphonic layers created on surface Ti-DLC. Each bar represents the average and SD taken from *n* ≥ 3 coatings (plates) from three independent experiments. The comparison was made using one-way analysis of the Student *t*-test. * *p* ˂ 0.05 vs. control group.

**Table 1 materials-12-02365-t001:** Results of wettability measurements of Ti-DLC coatings before and after modification by DP and PFDP components.

Sample	SCA (Deg)	θ_a_ (Deg)	θ_r_ (Deg)	Δθ (θ_a_ − θ_r_; Deg)
Ti-DLC1	74.6 ± 0.6	83.0 ± 2.1	45.1 ± 1.3	37.9 ± 0.9
Ti-DLC2	72.2 ± 0.8	78.7 ± 1.9	32.4 ± 0.8	46.3 ± 1.6
Ti-DLC3	64.8 ± 2.6	67.3 ± 2.2	15.1 ± 0.4	52.0 ± 1.3
Ti-DLC1/DP	115.2 ± 1.3	128.9 ± 3.2	121.2 ± 3.0	7.7 ± 0.2
Ti-DLC2/DP	114.2 ± 0.7	121.4 ± 3.0	108.9 ± 2.7	12.4 ± 0.3
Ti-DLC3/DP	104.6 ± 1.5	123.1 ± 3.1	104.1 ± 2.6	19.0 ± 0.5
Ti-DLC1/PFDP	127.3 ± 0.4	129.1 ± 3.3	122.8 ± 3.0	6.3 ± 0.2
Ti-DLC2/PFDP	126.6 ± 0.4	126.7 ± 3.1	117.2 ± 2.9	9.6 ± 0.3
Ti-DLC3/PFDP	123.9 ± 1.0	125.3 ± 3.1	111.0 ± 2.7	14.4 ± 0.4

**Table 2 materials-12-02365-t002:** Surface free energy and their components estimated for different Ti-DLC coatings with and without perfluoro and alkylphosphonic self-assembled monolayers.

Sample	γ^TOT^ (mJ/m^2^)	γ^LW^ (mJ/m^2^)	γ^AB^ (mJ/m^2^)	γ_AB_^+^ (mJ/m^2^)	γ_AB_^−^ (mJ/m^2^)
Ti-DLC1	47.0 ± 1.8	38.4 ± 1.6	8.6 ± 2.6	0.1 ± 0.0	8.5 ± 0.4
Ti-DLC2	49.4 ± 1.7	40.3 ± 2.5	9.1 ± 3.0	0.3 ± 0.1	8.8 ± 0.3
Ti-DLC3	54.4 ± 2.9	41.1 ± 1.1	13.3 ± 2.6	0.5 ± 0.1	12.8 ± 0.7
Ti-DLC1/DP	32.7 ± 1.6	29.9 ± 2.1	2.8 ± 2.3	0.5 ± 0.1	2.3 ± 0.1
Ti-DLC2/DP	35.0 ± 2.6	31.9 ± 4.2	3.1 ± 2.9	0.6 ± 0.1	2.5 ± 0.1
Ti-DLC3/DP	36.2 ± 1.8	32.2 ± 2.3	4.0 ± 1.5	0.7 ± 0.1	3.3 ± 0.2
Ti-DLC1/PFDP	8.1 ± 2.5	5.1 ± 0.7	3.0 ± 2.7	0.6 ± 0.1	2.4 ± 0.1
Ti-DLC2/PFDP	9.0 ± 2.3	5.7 ± 0.4	3.3 ± 2.3	0.8 ± 0.1	2.5 ± 0.1
Ti-DLC3/PFDP	10.7 ± 1.7	6.8 ± 0.4	3.9 ± 1.7	1.0 ± 0.1	2.9 ± 0.1

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
