# Peer review of "Impact of Perfluoro and Alkylphosphonic Self-Assembled Monolayers on Tribological and Antimicrobial Properties of Ti-DLC Coatings"

_materials, 2019, doi:10.3390/ma12152365_

Round 1

Reviewer 1 Report

In order for readers, who will be interested in your results, to trace and reproduce the results, characterization of Ti-DLC should be more carefully conducted; for example, Raman spectrum (indicating carbon network structure), hardness, are thickness should be added. If possible, hydrogen content and XPS analysis (indicating surface chemical structure) is required to be shown.

Reviewer 2 Report

The paper is well-written but I have some questions and recommendations for changes:

1. Why is titanium (Ti) used as a doping agent? And why is such a concentration used?

2. In «Materials and methods» part the authors write "Titanium-containing DLC coatings with the thickness of (100±2) nm...». What method and device was used to control the thickness of the coating?

3. Do the presented in tables 1 and 2 data have statistically significant differences? This data should be processed using statistical analysis methods (for example as data in Figure 6).

4. In this article given the large number of references to current literature sources. But there is no information about:

- impact of -CH3 and -CF3 groups on the hydrophobicity of Ti-DLC coatings;

- how pure DLC coatings affect on antibacterial properties?

5. Are there any significant differences between the Ti-DLC1, Ti-DLC2 and Ti-DLC3 groups when RMS values were calculated?

6. Do titanium and its compounds have antimicrobial properties?
